# What matters to medical ward patients, and do we measure it? A qualitative comparison of patient priorities and current practice in quality measurement, on UK NHS medical wards

Samuel Pannick,[1,2] Stephanie Archer,[1] Susannah Jane Long,[1,3] Fran Husson,[1] Thanos Athanasiou,[4] Nick Sevdalis[5]

¹NIHR Imperial Patient Safety Translational Research Centre, Imperial College London, London, UK
²Gastroenterology, Imperial College Healthcare NHS Trust, London, UK
³Medicine for the Elderly, Imperial College Healthcare NHS Trust, London, UK
⁴Surgery and Cancer, Imperial College London, London, UK
⁵Centre for Implementation Science, Kings College London, London, UK

**Correspondence to**
Dr Samuel Pannick;
sam.pannick@nhs.net

## ABSTRACT

**Objectives** To compare the quality metrics selected for public display on medical wards to patients' and carers' expressed quality priorities.

**Methods** Multimodal qualitative evaluation of general medical wards and semi-structured interviews.

**Setting** UK tertiary National Health Service (public) hospital.

**Participants** Fourteen patients and carers on acute medical wards and geriatric wards.

**Results** Quality metrics on public display evaluated hand hygiene, hospital-acquired infections, nurse staffing, pressure ulcers, falls and patient feedback. The intended audience for these metrics was unclear, and the displays gave no indication as to whether performance was improving or worsening. Interviews identified three perceived key components of high-quality ward care: communication, staff attitudes and hygiene. These aligned poorly with the priorities on display. Suboptimal performance reporting had the potential to reduce patients' trust in their medical teams. More philosophically, patients' and carers' ongoing experiences of care would override any other evaluation, and they felt little need for measures relating to previous performance. The display of performance reports only served to emphasise patients' and carers' lack of control in this inpatient setting.

**Conclusions** There is a gap between general medical inpatients' care priorities and the aspects of care that are publicly reported. Patients and carers do not act as 'informed choosers' of healthcare in the inpatient setting, and tokenistic quality measurement may have unintended consequences.

## INTRODUCTION

Patient involvement is a priority for the patient safety and healthcare quality movement,[1] but how best to involve patients remains unclear. Policymakers favour the transparent publication of quality metrics (ie, performance reporting) as a means of engaging patients in their care, framing this engagement as an informed choice of healthcare provider.

### Strengths and limitations of this study

► Participants included older, frail patients and those who did not speak English as a first language—demographics often excluded from safety and quality research.

► Our results build on the findings from postdischarge survey studies, free from recall bias.

► Current inpatients are in a vulnerable position and this may have affected some of their interview responses.

► The findings of this single-site study may not be generalisable, although the 'static' performance measures seen at the study site are typical of those reported in other literature.

► We focused on ward displays; other repositories for quality metrics, not in public view, may better approximate patient priorities.

In the right context, providing appropriate information can improve patients' and carers' participation in their care, perhaps even improving outcomes.[2] In the UK, this 'informed choice' argument has led to the mandatory display of performance metrics on NHS inpatient wards.[3]

In the acute setting, however, inpatients are unlikely to use performance measures as would typical 'consumers'. These patients are rarely given a choice of provider. Instead, they are assigned to an available ward, or medical team, as determined by organisational capacity. In addition, the debilitation and stress of an acute illness can impede information processing, and intense anxiety can lead to active information avoidance. This may equally affect patients' families or carers, who focus on the immediate health concerns of their loved ones. Even when patients are comfortable accessing complex

information at home, they should be treated as 'situationally impaired' in the hospital environment.[4] Whether inpatients value service-level metrics, and how they relate to them, have not yet been evaluated.

Here, we compare the quality metrics selected for public display in NHS medical wards to patients' and carers' expressed quality priorities. We sought to capture patients' and carers' perceptions of a 'good ward', to better understand their reactions to the quality metrics on display.

## METHODS

The study was conducted on general medical wards, which provide the majority of acute inpatient care but struggle for organisational attention or targeted improvement strategies.[5] We assessed ward information displays in two acute medical wards and two geriatric wards at a tertiary NHS (public) hospital in London, with a proforma. This captured the type of performance metrics on public view (eg, specific hospital-acquired infections or pressure ulcers); whether the metrics themselves were clearly defined; whether there was a reference performance benchmark or goal; and how the information was displayed. Free text notes highlighted any adjacent information on the display boards.

Examples were photographed with a digital camera. These images provided insights into the time and priority that the displays were afforded in practice: images are powerful conduits for the feel and texture of environments.[6] Visual materials '*reveal what is hidden in the inner mechanisms of the ordinary*', providing perspective on everyday practices.[6] This 'visual sociology', or visual research method, also allows the researcher to reflect on what they encounter in their fieldwork. In doing so, photographic meaning is constructed: one must be aware that photos are not themselves unmediated or unbiased, but dependent on the viewer.[7] Although these documents of record are not undisputed, their value lies in triangulation with other data, in this case the objective categorisation of their contents, and in their interpretation by patients and carers.

The photos were used as prompts in semi-structured interviews with general medical inpatients and their carers at the hospital. Interviews are key tools 'in assessing user views of services and healthcare provision, and in revealing why some care is perceived as poor quality'.[8] The interviews were based on a topic guide, codeveloped with patient and carer representatives, exploring care priorities and the concept of a good ward (see online supplementary file). The topic guide was used flexibly, harnessing broad prompts and follow-up questions, in view of the different roles of the participants (carers and patients) and their varying lengths of hospitalisation.

Ward staff (doctors, nurses and allied professionals) were asked to suggest patients or carers who would be physically capable of taking part in an interview. Participants were aged over 18 years, and able to provide informed consent. Patients were excluded if they were physiologically unstable, had major cognitive or communication difficulties or did not speak English. The interviews took place at patients' bedsides, as described previously in qualitative work with hospitalised medical patients.[9] Unintentional power relationships and a false therapeutic rapport can develop within sensitive interviews, with implications for data quality.[10 11] To mitigate this, no members of the study team were involved in the participants' clinical care, and this was clearly communicated to the interviewees when they gave their consent to take part in the study. In addition, the interviews were framed as entirely separate from their ongoing clinical care.[11] The interviews were conducted by a specialist registrar in internal medicine and gastroenterology, undertaking a PhD in healthcare quality improvement, with previous experience of qualitative research (SP). Interviews were audiotaped, and then transcribed verbatim. Using NVivo (QSR International, Australia), two researchers trained in qualitative methods (SP—doctor and SA—psychologist) analysed the transcripts using an inductive (theory-generating) thematic analysis.[12] Each researcher coded the transcripts individually, generating an individual coding frame, which was then discussed and refined between the two coders. The transcripts were coded again, before a group of higher order themes was proposed. A third round of analysis—individually, and then with consensus—confirmed these metathemes and the aggregation of coded transcript fragments within them. The two researchers serially reviewed these results as the interviews were ongoing, and data collection ceased when the study reached saturation, that is, when no new themes were becoming apparent.

## PATIENT AND PUBLIC INVOLVEMENT

The interview topic guide was coproduced with local patient and carer representatives, who in turn canvassed their patient and carer networks for opinions and feedback. The patient representative (FH) coauthored the final manuscript reporting the study's results.

## RESULTS

### Interview participants

Fourteen people were interviewed (nine patients and five carers). Seven people (four carers) were female. Patients had a median age of 75 (range 57–86), with a median length of stay of 5 days. Seventy-one per cent of participants spoke English as a first language. Fourty-four per cent (4/9) of patients depended on family or community support, and 33% (3/9) of them had undergone other hospital admissions in the preceding 6 months. Nine interviews took place on the acute medical wards and five on medicine for the elderly wards. Interviews lasted a median of 23 min (range 11–48 min).

**Table 1** Performance indicators available in public ward areas

| Performance indicator | Display choice | Display format |
|---|---|---|
| Hand hygiene | Percentage in last audit | Most recent result only; numerator and denominator definitions not provided. |
| Hospital-acquired infections | Date of last recorded event | Most recent result only |
| Pressure ulcers | Date of last recorded event | Most recent result only |
| Falls | Date of last recorded event | Most recent result only |
| Nurse staffing | Numbers of staff required for the shift versus those actually on duty, for staff nurses and healthcare assistants | Most recent result only; explanation of staff responsibilities |
| Patient feedback | 'Friends and Family Test' star rating; percentage of patients who would recommend the ward* | Most recent result only; examples of patients' comments; no explanation of star rating system |

*The 'Friends and Family Test' asks 'How likely are you to recommend our service to friends and family if they needed similar care or treatment?'[28]

### What performance metrics were on display, and how were they portrayed?

Performance metrics evaluated hand hygiene, hospital-acquired infections (meticillin-resistant *Staphylococcus aureus* and *Clostridium difficile*), nurse staffing, pressure ulcers, falls and patient feedback (see online supplementary file and table 1). The intended audience for these metrics was often unclear: individual display boards contained combinations of messages for patients and staff. Possessive pronouns (our and your) and pronouns (we and you) were used interchangeably, within the same display, to refer to both patients and staff.

Performance measures were displayed with little background information or context. Each metric was displayed as a single, static measure of performance, with no evidence of trends over time. There was no indication of an acceptable benchmark. No patient-actionable information was given for any of the performance measures, other than a suggestion to speak to a senior nurse for more information about staffing on the ward. Ward displays about local quality and safety priorities (eg, 'MRSA compliance') were not explicitly linked to previous performance.

Patient and carer interviews were wide ranging. For ease of understanding, we have aggregated the results into the following sections.

### What makes a good ward in the eyes of the patients and their carers?

The interviews identified three key components of high-quality ward care: communication, staff attitudes and hygiene.

### Communication

Participants felt entirely dependent on staff to keep them abreast of forthcoming investigations and treatments. They valued prompt communication and were keenly aware of its absence. At the same time, they recognised that treatment plans would frequently change, often for reasons outside of their teams' control and simply held those teams accountable for keeping them updated:

> I know it is not always possible that definitive information is available. But as long as you are informed to the ability that they can inform you, you cannot have any gripes about that. If someone says to you, 'Look, you may go home tomorrow', I am big enough and ugly enough to know that it may be the day afterwards… (Patient 3)

The value of effective, shared communication within the multidisciplinary team was also highlighted. The capacity to speak to one team member, and have that conversation disseminated promptly to the rest of the team, was a key feature of good performance:

> I have found you'll be speaking to one person—and it could be a nurse or a doctor or anybody else—and at the end of the day, everybody knows what I'm talking about… So you can communicate with [just] one person… It's a vital thing. (Carer 1)

Most comments about information sharing within multidisciplinary teams came from carers, rather than patients. This perhaps reflected the role of carers in the ward environment, where they act both as an information source for professional teams and as advocates for the patients.

### Staff attitudes

The second element of high-quality care was staff attitude. Considerable attention was paid to *how* staff went about their work: staff attentiveness, or 'service', influenced whether patients felt they were on a good ward. Adjectives like 'jolly', 'respectful' and 'helpful', or 'abrupt' and 'wishy-washy', were not so much seen as individual personality attributes, as they were features of work performance:

> [A] good ward is to be helpful to patients, being more human than a machine, you understand? (Patient 2)

I think it's the attitude of people [that makes a good ward]. It's the main thing. (Patient 6)

Thus, the manner of care delivery—rather than the resources available for it—largely defined the care experience. The corollary of this was the potential for a major change between one shift and the next, even on the same ward. There was a sense, perhaps, that rather than a good or bad ward there were just good shifts or bad shifts:

Where it changes more than anything else is at night, when you have a complete change of staff. Sometimes the night staff that come on are absolutely fantastic, and are very engaged. But sometimes they are entirely the opposite. It is like, 'Well we are just here to get you through until the morning, when the people that are looking after you come back.' (Patient 3)

As well as analysing their own interactions with staff, patients and carers were keen observers of the working relationships between different professionals on the ward. Whether staff seemed appreciative of each other's efforts, or were openly disrespectful to one another, caused patients to wonder how they too were being treated:

[You might think] the more staff, the better the person feels, and that is not how I feel… Everything depends on the lower level[s] of staff we've got working in the ward… and their position [should be] respected by the doctors and the more senior people… They did all [sorts of tasks], and nobody seemed to recognise that they were doing something like that… (Patient 7)

Two nurses were having a fight with each other, and that's not very good for the rest of us. And of the course the supervisor was asking them to be quiet, because they were shouting and screaming at each other. (Patient 4)

Yet, these observations of staff behaviours were quite nuanced. Patients recognised different types of unproductive working relationships, describing over-familiarity ('*almost like a bunch of friends working together*'—Patient 3), as well as open antagonism. They also made allowances for the general workload on the ward, even excusing displays of inappropriate behaviour:

There's a lot of pressure put on the staff, you know it's understandable. You can see that they're actually very tired people, they needed a good rest, and that's why the whole thing gets on top of them, they're overworked. (Patient 4)

### Hygiene
In a similar vein, patients and carers expressed quite subtle views of why they held hygiene standards to be important. First, good hygiene was *de facto* evidence of a ward that was providing safe care, with little risk of iatrogenic infection. Patients and carers were conscious of the possibility of hospital-acquired infection, understanding it as a major

risk associated with inpatient care. Minimising that risk made it possible to focus on the acute medical issues at hand. Second, good hygiene served as a deeper marker of staff pride, diligence and attention to detail, all of which were reassuring:

The cleanliness aspect, I think, is… more important than possibly people realise… It sets out a marker if you like… if the mindset of the ward is, you know, 'We are proud of the place that we work in.' So it is a fairly good marker of how that ward will actually be. (Patient 3)

### How did patients and carers perceive the quality metrics on display?
#### Benefits—using infection data for hand hygiene, and understanding staff performance
Patients and carers described some benefits of the quality metrics on display, particularly when it came to infection data. They acknowledged prompts to focus on their own hand hygiene, while hoping that staff would do the same. In some cases, a vague familiarity with infection control terminology was helpful:

Because that's in the press [MRSA rates], I suppose people do want to know that, don't they? All of this you read in the papers of people being in hospital—they went in with one thing and they came out with that… You don't want to get worse. They're meant to be making you better. (Patient 5)

Real-time information on staffing levels was also potentially helpful, in that it could help set realistic expectations of the care patients might receive:

When I'm getting poor service on a particular day, at least I can see that there might be a good reason for it… I would be more understanding, if I had to wait twice as long for help, if I knew that there was only half the number of staff there should be. (Patient 8)

When I saw the amount of staff that you're supposed to have on the ward, there were not half the staff. So the other staff that turned up were constantly busy, running back and forth, and you can see how much stress they were [under]. But they were doing a good job… You can see the nurse who has turned up is doing a really good job. (Carer 1)

#### Significant drawbacks—problems in information delivery, prioritisation of personal experience and unintended consequences
However, patients and carers were largely disparaging about the quality metrics on display. There were numerous problems with information delivery, such as inadequate font size or colour contrast. Yet even with these issues addressed, the information provided was fundamentally inadequate to make a judgement about quality. Patients struggled to see the relevance of a single figure when no trend or benchmark was provided:

Obviously as a member of the public I want the minimum [information], but I have nothing to compare it with. So if you [say], 'We've not had one [infection] for three years', I can't compare that with anything. So it doesn't mean anything to me… (Carer 5)

That [single figure] doesn't mean anything. That doesn't inform. It could be an increase… but it could be [a] decrease. (Patient 2)

More broadly, participants felt little need for measures that related to their wards' previous performance. Their ongoing experiences of care would override any other evaluation. From each individual's perspective, their personal care was the priority, whether or not it reflected a typical standard of care on that ward. In that light, other performance metrics became irrelevant:

I use my own judgement. If I'm satisfied: that's it. (Patient 9)

If we want information, we ask for it and we get it. As long as [my relative] is alright and getting looked after, I'm not really bothered about nothing else. If she's getting well looked after, the nurses are lovely, their care is great… that's all we are concerned about. (Carer 3)

As a result, the production of ward quality metrics had some unintended consequences, even going so far as to reduce patients' trust in the whole enterprise. The absence of baseline data in quality displays in particular raised suspicions that poor performance was being concealed.

Let us face it… you have got your 100% figure there. Would you put up a 20% figure?… What would you be doing? You would be ruining the confidence of the patients… (Patient 3)

Patients and carers felt that staff had to have ownership of the quality agenda in hospital: quality metrics were for staff—not patients—to digest. Many interviewees drew comparisons with other settings in which they were consumers: as restaurant diners, or as car purchasers, where their ability to exercise a choice was crucial. Here, however, they had no power to choose, and the display of performance reports only served to emphasise their lack of control:

It would be great if I'm admitted and I'm given a choice of five wards, and I would say, 'Well, how do I know which one's which, which one's best?' My next question would be, 'Can you give me the audits of those wards to show which has the highest rating?' and I would go to that… If there's no choice, then it's all academic. (Patient 8)

Well, other than clean the wards, there's not a lot we can do is there? What else can you do? (Carer 4)

The 'Friends and Family Test' question was found to be particularly challenging, given that these patients had no choice in arriving on the ward in the first place, nor could subsequent patients exercise a preference to get there. Indeed, the service pressures on hospital admissions were so well publicised that the idea of choosing a ward seemed faintly ridiculous:

Would I recommend a ward? How can you recommend a ward?… I mean, that's a daft question, because… they put you in the place you need to be, don't they? (Patient 5)

## DISCUSSION

To our knowledge, this is the first study to compare publicly displayed performance metrics with patient and carer perceptions of high-quality care on UK medical wards. We identified discrepancies between patient-identified and carer-identified priorities and the quality metrics relating to their care on general medical wards. Patients and carers expressed three core components of high-quality general medical care: communication, staff attitudes and hygiene. These were only partially aligned with the performance measures on display. Specifically, we found process and outcome measures relating to hand hygiene and iatrogenic infection, but none specifically relating to attitudes or communication. Patients and carers acknowledged limited benefits to the display of performance data, but had significant reservations about how it was contextualised. They relied on their own experience of care to judge its quality, above any objective measure of performance. More philosophically, they questioned the purpose of publicly displayed performance data, given their lack of choice in this setting. In some cases, these reservations actually eroded trust in ward teams' performance.

This study builds on a body of research exploring patient priorities and patient involvement in the acute hospital setting. Boyd surveyed recently discharged patients, similarly finding that communication, patient–professional interactions, hygiene and the technical delivery of care were their main priorities.[13] Our study suggests that Boyd's findings (which excluded current inpatients) were not unduly affected by recall bias. Nonetheless, hospitalised patients remain relatively indifferent to service-level performance and change.[14] We suggest an explanation for this: current inpatients are unable to exercise informed choices about their ward, nor are they able to directly use information to improve performance. They are therefore excluded from the two key pathways by which performance measurement may lead to quality improvement.[15]

Our findings question the mandatory collection, and display, of performance data that do not align with patient priorities. These data collection exercises have considerable opportunity costs. We note the recent call for the abolition of the mandatory Friends and Family Test, one of the performance indicators we found on display, which has been criticised on similar lines.[16] These

data sets are expensive to maintain, 'at best tolerated, often ignored, and sometimes ridiculed'.[16] The resulting tokenistic display of performance data erodes patients' trust in the system that organises and governs their care. It can also be corrosive for staff morale, both at the frontline and at board level.[17 18] This tokenism is perpetuated by a dearth of resources for implementing meaningful improvement.[1 19] A credible, coproduced, quality framework for acute medical inpatients is urgently required, with outcomes that are sensitive to the work[20] and structures[21] of inpatient care. Coproduced quality standards should capitalise on the active contributions of patients and carers, rather than depicting them as 'informed choosers' of healthcare provision.

Study limitations include a relatively small sample from a single site. We collected demographic data for patients but not their carers. Nonetheless, the group of interviewees included demographics often excluded from safety and quality research: older, frail patients and those who do not speak English as a first language.[22] The study reached data saturation, with no new themes emerging as the final interviews took place. Ward displays at this site were also typical for regional practice. 'Static' performance measures, as seen here, are widespread, and even the performance data presented at healthcare board level rarely depicts the role of chance in the formation of data patterns.[23 24] Finally, other repositories for quality metrics, beyond those ward displays analysed here, may better approximate patient priorities. However, they typically use composites of the data we found,[25] or are aggregated to the hospital level, with no ward-level interpretation.[26 27]

In conclusion, we found a gap between general medical inpatients' care priorities and the aspects of care that are publicly reported. Where performance measurement could have been useful to patients and carers, suboptimal displays only served to emphasise their passive receipt of services. Unless patients and carers are invited to define the quality metrics they hold relevant, ward services may struggle to engage them in improvement efforts. Ultimately, tokenistic quality measurement may have unintended consequences, eroding patients' trust in their healthcare teams.

**Acknowledgements** We are grateful to Drs Louise Hull and Tayana Soukup for their help in facilitating the focus groups. We thank Ms Margaret Turley (carer representative) from the NIHR Imperial PSTRC for her help developing the interview guide.

**Contributors** Study design: SP, SA, TA, FH, NS and SJL. Study implementation and data collection: SA, SP and SJL. Analysis: SA, SP, NS, FH and TA. All authors contributed to, read and approved the final manuscript.

**Funding** This paper represents independent research supported by the National Institute for Health Research (NIHR) Imperial Patient Safety Translational Research Centre, and the Imperial College Healthcare Charity (Grant GG14\1022). NS's research is supported by the NIHR Collaboration for Leadership in Applied Health Research and Care South London at King's College Hospital NHS Foundation Trust. NS is a member of King's Improvement Science, which is part of the NIHR CLAHRC South London and comprises a specialist team of improvement scientists and senior researchers based at King's College London.

**Disclaimer** The views expressed are those of the authors and not necessarily those of the NHS, the NIHR or the Department of Health and Social Care.

**Competing interests** None declared.

**Patient consent for publication** Not required.

**Ethics approval** Ethical approval was granted by the Westminster Research Ethics Committee (16/LO/0196) and the hospital's joint research compliance office (16SM3129).

**Provenance and peer review** Not commissioned; externally peer reviewed.

**Data sharing statement** There are no additional unpublished data.

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
