## [Reviewer comments · BMJ Open]

BMJ Open

BMJ Open is committed to open peer review. As part of this commitment we make the peer review history of every article we publish publicly available.

When an article is published we post the peer reviewers' comments and the authors' responses online. We also post the versions of the paper that were used during peer review. These are the versions that the peer review comments apply to.

The versions of the paper that follow are the versions that were submitted during the peer review process. They are not the versions of record or the final published versions. They should not be cited or distributed as the published version of this manuscript.

BMJ Open is an open access journal and the full, final, typeset and author-corrected version of record of the manuscript is available on our site with no access controls, subscription charges or pay-per-view fees (<http://bmjopen.bmj.com>).

If you have any questions on BMJ Open's open peer review process please email info.bmjopen@bmj.com

BMJ Open

What do medical ward patients want, and do they get it? A comparison of patient priorities in care quality and current practice in quality measurement

Journal:	BMJ Open
Manuscript ID	bmjopen-2018-024058
Article Type:	Research
Date Submitted by the Author:	08-May-2018
Complete List of Authors:	Pannick, Samuel; Charing Cross Hospital, Gastroenterology Archer, Stephanie; Imperial College London, NIHR Imperial Patient Safety Translational Research Centre; Imperial College London, Long, Susannah; Imperial College London, Centre for Patient Safety and Service Quality Husson, Fran; Imperial College, Faculty of Medicine Athanasίου, Thanos; Imperial College London, Surgery and Cancer Sevdalis, Nick; King's College London,
Keywords:	Quality in health care < HEALTH SERVICES ADMINISTRATION & MANAGEMENT, GENERAL MEDICINE (see Internal Medicine), Organisation of health services < HEALTH SERVICES ADMINISTRATION & MANAGEMENT, Clinical governance < HEALTH SERVICES ADMINISTRATION & MANAGEMENT

Only

**What do medical ward patients want, and do they get it?**
**A comparison of patient priorities in care quality and current**
**practice in quality measurement**

Samuel Pannick,^{1,2} Stephanie Archer,¹ Susannah J Long,^{1,2} Fran Husson,^{1,2}
Thanos Athanasiou,³ & Nick Sevdalis⁴

¹ NIHR Imperial Patient Safety Translational Research Centre, Imperial College London, UK

² Imperial College Healthcare NHS Trust, London, UK

³ Department of Surgery & Cancer, Imperial College London, UK

⁴ Centre for Implementation Science, King's College London, UK

Corresponding author:

Dr Sam Pannick, MA MB BS MRCP PhD

Specialist Registrar in Gastroenterology and Internal Medicine, Department of

Gastroenterology, Charing Cross Hospital, Fulham Palace Road, W6 8RF

United Kingdom

sam.pannick@nhs.net; + (44) 203 311 1234

*Author information:*

Dr Stephanie Archer, Research Fellow, NIHR Imperial Patient Safety Translational Research
Centre, Imperial College London.

Dr Susannah J Long, Consultant Geriatrician, Imperial College Healthcare NHS Trust, and
Honorary Clinical Senior Lecturer, NIHR Imperial Patient Safety Translational Research
Centre, Imperial College London.

Ms Fran Husson, Patient Advocate, Imperial College Healthcare NHS Trust, and Honorary
Research Officer, Imperial College London.

Professor Thanos Athanasiou, Consultant Cardiothoracic Surgeon, Imperial College
Healthcare NHS Trust, and Professor of Cardiovascular Sciences and Cardiac Surgery,
Imperial College London.

Professor Nick Sevdalis, Professor of Implementation Science and Patient Safety, King's
College London.

*Word count: 3133*

*Conflicts of interest:* This paper represents independent research supported by the National
Institute for Health Research (NIHR) Imperial Patient Safety Translational Research Centre,
and the Imperial College Healthcare Charity (Grant GG14\1022). NS' research is supported
by the NIHR Collaboration for Leadership in Applied Health Research and Care South London
at King's College Hospital NHS Foundation Trust. NS is a member of King's Improvement
Science, which is part of the NIHR CLAHRC South London and comprises a specialist team of
improvement scientists and senior researchers based at King's College London. Its work is
funded by King's Health Partners (Guy's and St Thomas' NHS Foundation Trust, King's
College Hospital NHS Foundation Trust, King's College London and South London and

Maudsley NHS Foundation Trust), Guy's and St Thomas' Charity, the Maudsley Charity and
the Health Foundation. No funding source had any role in the design and conduct of the
study; collection, management, analysis or interpretation of the data; or preparation, review
or approval of the manuscript. The views expressed are those of the authors and not
necessarily those of the NHS, the NIHR or the Department of Health and Social Care.

*Contributions:*

Study design: SP, SA, TA, FH, NS, SJL. Study implementation and data collection: SA, SP, SJL.
Analysis: SA, SP, NS, FH, TA. All authors contributed to, read and approved the final
manuscript.

*Ethical approval:*

Ethical approval was granted by the Westminster Research Ethics Committee (16/LO/0196)
and the hospital's joint research compliance office (16SM3129).

*Acknowledgements:* We are grateful to Drs Louise Hull and Tayana Soukup for their help in
facilitating the focus groups. We thank Ms Margaret Turley (carer representative) from the
NIHR Imperial PSTRC for her help developing the interview guide.

*Data sharing:* There are no additional unpublished data.

*Key words:* healthcare quality; medical ward; patient experience

ABSTRACT

Objectives

To compare the quality metrics selected for public display in NHS medical wards to patients' and carers' expressed quality priorities.

Design

Observational assessment of general medical ward practice and semi-structured interviews.

Setting

UK tertiary hospital

Participants

Fourteen patients and carers on acute medical wards and geriatric wards.

Results

Quality metrics on public display evaluated hand hygiene, hospital-acquired infections, nurse staffing, pressure ulcers, falls, and patient feedback. The intended audience for these metrics was unclear, and the displays gave no indication as to whether performance was improving or worsening. Interviews identified three perceived key components of high quality ward care: communication, staff attitudes, and hygiene. These aligned poorly with the priorities on display. Incomplete performance reporting had the potential to reduce patients' trust in their medical teams. More philosophically, patients' and carers' ongoing experiences of care would override any other evaluation, and they felt little need for measures relating to previous performance. The display of performance reports only served to emphasise patients' and carers' lack of control in this inpatient setting.

Conclusions

There is a persistent gap between general medical inpatients' care priorities and the aspects of care that are publicly reported. Patients and carers do not act as 'informed choosers' of healthcare in the inpatient setting, and tokenistic quality measurement may have unintended consequences.

STRENGTHS AND LIMITATIONS OF THE STUDY

- The study highlights the differences between inpatients' views of care quality and the care priorities expressed through public performance reporting.
- Participants included older, frail patients, and those who did not speak English as a first language – demographics often excluded from safety and quality research.
- The 'static' performance measures seen at the study site are typical of those reported in other literature.
- Other repositories for quality metrics, beyond the ward displays analysed here, may better approximate patient priorities.

INTRODUCTION

Patient involvement is a priority for the patient safety and healthcare quality movement,¹ but how best to involve patients remains unclear. Policymakers favour the transparent publication of quality metrics as a means of engaging patients in their care, framing this engagement as an informed choice of healthcare provider. In the right context, providing appropriate information can improve patients' and carers' participation in their care, perhaps even improving outcomes.² In the UK, this 'informed choice' argument has led to the mandatory display of performance metrics on NHS inpatient wards.³

In the acute setting, however, inpatients are unlikely to use performance measures as would
typical 'consumers'. Few choices are available. The debilitation and stress of an acute illness
can impede information processing, and intense anxiety can lead to active information
avoidance. This may equally affect patients' families or carers, who focus on the immediate
health concerns of their loved ones. Even when patients are comfortable accessing complex
information at home, they should be treated as 'situationally-impaired' in the hospital
environment.⁴ Whether inpatients value service-level metrics, and how they relate to them,
have not yet been evaluated.

Here, we compare the quality metrics selected for public display in NHS medical wards to
patients' and carers' expressed quality priorities. We aim to capture patients' and carers'
perceptions of a 'good ward', and evaluate their reactions to the quality metrics on display.
Our secondary aim was to identify a set of quality metrics which might align incentives for
the varied stakeholders on these units – including staff, managers and patients.

**METHODS**

The study was conducted on general medical wards, which provide the majority of acute
inpatient care but struggle for organisational attention or targeted improvement strategies.⁵
We assessed ward information displays in acute medical wards and geriatric wards at a
tertiary hospital in London, with a standardised instrument. Free text notes highlighted any
adjacent information on the display boards. Examples were photographed with a digital
camera. Photos were then used as prompts in semi-structured interviews with general
medical inpatients and their carers at the hospital. The interviews were based on a topic
guide, co-developed with patient and carer representatives, exploring care priorities and the
concept of a 'good ward' [**Online supplement**].

Ward staff (doctors, nurses, and allied professionals) were asked to suggest patients or
carers who would be physically capable of taking part in an interview. Participants were
aged over 18 years, and able to provide informed consent. Patients were excluded if they
were physiologically unstable, had major cognitive or communication difficulties, or did not
speak English. The interviews took place at patients' bedsides, as described previously in
qualitative work with hospitalised medical patients.⁶ The interviews were conducted by a
specialist registrar in internal medicine and gastroenterology, undertaking a PhD in
healthcare quality improvement, with previous experience of qualitative research (SP).
Interviews were audiotaped, and then transcribed verbatim. Using NVivo (QSR International,
Australia), two researchers (SP and SA) analysed the transcripts using an inductive thematic
analysis.⁷ Data collection ceased when the study reached saturation, with no new themes
emerging. Other qualitative interview studies reached data saturation within the first 12
interviews.⁸

[revised manuscript text omitted]

of interviewees was a representative one, and the study reached data saturation.

Participants included older, frail patients, and a significant proportion did not speak English
as a first language – demographics often excluded from safety and quality research.¹⁹ Ward
displays were also representative: the use of 'static' performance measures, as seen here, is
widespread.²⁰ There are other repositories for quality metrics, beyond those ward displays
analysed here, which may better approximate patient priorities. However, they typically use
composites of the data we found,²¹ or are aggregated to the hospital level, with no ward-
level interpretation.^{22,23}

**CONCLUSION**

There is a persistent gap between general medical inpatients' care priorities and the aspects
of care that are publicly reported. Tokenistic quality measurement may have unintended
consequences, eroding patients' trust in ward teams.

REFERENCES

1. Lawton R, O'Hara JK, Sheard L, et al. Can patient involvement improve patient safety? A cluster randomised control trial of the Patient Reporting and Action for a Safe Environment (PRASE) intervention. *BMJ Qual Saf* 2017;**26**(8):622-31.
2. Shaller D, Kanouse DE, Schlesinger M. Context-based strategies for engaging consumers with public reports about health care providers. *Med Care Res Rev* 2014;**71**(5 Suppl):17S-37S.
3. Department of Health. Hospital 'friends and family test' announced. 2012. <https://www.gov.uk/government/news/hospital-friends-and-family-test-announced>. Accessed 11/12/2017.
4. Morris D, Karlson A. Dynamic Accessibility Requirements for Hospital Patients. SIGCHI Conference on Human Factors in Computing Systems. Vancouver, Canada: ACM Press, 2011.
5. Pannick S, Wachter RM, Vincent C, et al. Rethinking medical ward quality. *BMJ* 2016;**355**:i5417.
6. Tobiano G, Bucknall T, Marshall A, et al. Patients' perceptions of participation in nursing care on medical wards. *Scand J Caring Sci* 2016;**30**(2):260-70.
7. Braun V, Clarke V. Using thematic analysis in psychology. *Qualitative Research in Psychology* 2006;**3**(2):77-101.
8. Guest G, Bunce A, Johnson L. How Many Interviews Are Enough?: An Experiment with Data Saturation and Variability. *Field Methods* 2006;**18**(1):59-82.
9. NHS England. Friends and Family Test: Guidance. 2014. Publications Gateway Ref No. 01787.
10. Boyd J. The 2006 inpatients importance study. Oxford: Picker Institute, 2007. http://www.nhssurveys.org/Filestore/documents/Findings_and_development_of_the_2006_Inpatients_Importance_study_final.pdf.
11. Wray CM, Farnan JM, Arora VM, et al. A qualitative analysis of patients' experience with hospitalist service handovers. *J Hosp Med* 2016;**10**.1002/jhm.2608.
12. Berwick DM, James B, Coye MJ. Connections between quality measurement and improvement. *Med Care* 2003;**41**(1 Suppl):i30-8.
13. Robert G, Cornwell J, Black N. Friends and family test should no longer be mandatory. *BMJ* 2018;**360**:k367.
14. NHS England. Review of the Friends and Family Test. 2014. <https://www.england.nhs.uk/wp-content/uploads/2014/07/fft-rev1.pdf>. Accessed 16/10/2017.
15. Raleigh V, Thompson J, Jabbal J, et al. Patients' experience of using hospital services. London: The King's Fund, 2015. <http://www.pickereurope.org/wp-content/uploads/2015/12/Patients-experience-Kings-Fund-Dec-2015.pdf>.
16. Sheard L, Marsh C, O'Hara J, et al. The Patient Feedback Response Framework - Understanding why UK hospital staff find it difficult to make improvements based on patient feedback: A qualitative study. *Soc Sci Med* 2017;**178**:19-27.
17. Pannick S, Davis R, Ashrafian H, et al. Effects of Interdisciplinary Team Care Interventions on General Medical Wards: A Systematic Review. *JAMA Intern Med* 2015;**175**(8):1288-98.
18. Dumitrascu AG, Burton MC, Dawson NL, et al. Patient portal use and hospital outcomes. *Journal of the American Medical Informatics Association* 2018;**25**(4):447-53.

19. O'Hara JK, Lawton RJ. At a crossroads? Key challenges and future opportunities for patient involvement in patient safety. *BMJ Qual Saf* 2016;10.1136/bmjqs-2016-005476.
20. Anhoj J, Hellesoe AB. The problem with red, amber, green: the need to avoid distraction by random variation in organisational performance measures. *BMJ Qual Saf* 2016;10.1136/bmjqs-2015-004951.
21. NHS Quality Observatory. NHS Safety Thermometer. https://www.safetythermometer.nhs.uk/index.php?option=com_content&view=article&id=1&Itemid=101. Accessed 24th June 2016.
22. Care Quality Commission. NHS Surveys: focused on patients' experience. 2016. <http://www.nhssurveys.org/>. Accessed 23rd June 2016.
23. Centers for Medicare and Medicaid Services. HCAHPS Fact Sheet. June 2015. Baltimore, MD. <http://www.hcahpsonline.org/facts.aspx>. Accessed 23rd June 2016.

Table 1: Performance indicators available in public ward areas

Performance indicator	Display choice	Display format
Hand hygiene	Percentage in last audit	Most recent result only; numerator and denominator definitions not provided.
Hospital-acquired infections	Date of last recorded event	Most recent result only
Pressure ulcers	Date of last recorded event	Most recent result only
Falls	Date of last recorded event	Most recent result only
Nurse staffing	Numbers of staff required for the shift vs those actually on duty, for staff nurses and health care assistants	Most recent result only; explanation of staff responsibilities
Patient feedback	'Friends and Family Test' star rating; percentage of patients who would recommend the ward*	Most recent result only; examples of patients' comments; no explanation of star rating system

*The 'Friends and Family Test' asks "How likely are you to recommend our service to friends and family if they needed similar care or treatment?"⁹

TOPIC GUIDE: Current patient / carer for current patient**1. Let's find out about you.**

Age
Employment
Education – school / university / postgraduate
Social support structures & marital status
Ethnicity

2. Why are you in hospital now?

Current diagnosis
Other conditions
Approximate length of stay to date
 < 1 day
 1 – 5 days
 5 – 10 days
 > 10 days

3. How many times have you been admitted to hospital in the last 6 months?

1-5
5-10
>10

4. Do you always come to this hospital or have you been admitted to other local hospitals?**5. How do you know if you're on a good ward? What is a 'good ward' to you?**

Environment

- Clean
- Quiet
- Toilet and shower are available when required
- Meal timeliness, warmth
- Help available when requested
- Staff are responsiveness to my needs / my family's needs

Welcome

- My arrival is expected
- Staff introduce themselves
- Staff make me feel I will be well looked after; show a caring attitude; and don't rush me

Communication and use of personal information

- Accurate knowledge of previous medical history / current diagnosis / current investigations / discharge plan / medication reconciliation
- Quality of communication / teamwork

Discharge preparation

Friends' / families' recommendations

Ward information boards / quality and safety boards

Ward information leaflets / other printed materials.

Ward information displays / electronic screens

**6. If you had to decide whether a ward was good or not, what information would you need to**
**make that decision?**

**7. Have you noticed any of the information the ward displays about itself? What do you think of**
**the information you've seen?**

Friends and family test results

Safety cross

Shift-by-shift staffing

Falls

Pressure ulcers

Safety thermometer / harm-free care

Venous thromboembolism prophylaxis

Hand hygiene compliance

Hospital-acquired infections

Infection rates

Incident reporting

**8. What would you like to know about how your ward is performing?**

Hand hygiene compliance

Staffing levels

Friends and family results

Hospital-acquired infections

Pressure Ulcers

Falls

Venous thromboembolism prophylaxis

Complaints

Compliments

Length of stay

Mortality

Readmission rate

Safety climate

**9. How should your ward make that information available to you and your family??**

Ward displays

Leaflets

Smartphone / other device

Webpage

**10. Preference for information seeking**

Information-seeking sub-scale

	Disagree strongly	Disagree slightly	Neutral	Agree slightly	Agree strongly
11 12 As you become sicker you should be told more and more about your illness
13 14 15 You should understand completely what is happening inside your body as a result of your illness					
16 17 18 Even if the news is bad, you should be well informed					
19 20 21 Your doctor should explain the purpose of your laboratory tests					
22 23 24 It is important for you to know all the side effects of your medication					
25 26 27 Information about your illness is as important to you as treatment					
28 29 30 When there is more than one method to treat a problem, you should be told about each one					

**11. Have you previously had to complain about care or healthcare staff, nurses or doctors? What made you complain? How? PALS / informally / in writing?**

I-Staff Safe Staffing Information Board

The Nurse in Charge today is:

[Redacted]

Staffing Status

[Redacted]

DATE:

15-2-15

Current Shift

Registered Nurses

Staff required

On duty

3

1

Healthcare Assistants

2

2

Additional Support
(for e.g. specials)

Roles and Responsibilities

	Responsibilities
Nurse in Charge	Provides supervision, management and direct clinical care
Registered Nurses	Provides direct clinical care and supervision
Health Care Assistants	Provides direct care under the supervision of a registered nurse
Additional Support	Specials (may be registered or unregistered providing 121 direct care with supervision)

If you have any queries about the staffing on this ward please speak to the Nurse in Charge

COREQ (COnsolidated criteria for REporting Qualitative research) Checklist

A checklist of items that should be included in reports of qualitative research. You must report the page number in your manuscript where you consider each of the items listed in this checklist. If you have not included this information, either revise your manuscript accordingly before submitting or note N/A.

Topic	Item No.	Guide Questions/Description	Reported on Page No.
Domain 1: Research team and reflexivity			
Personal characteristics			
Interviewer/facilitator	1	Which author/s conducted the interview or focus group?	
Credentials	2	What were the researcher's credentials? E.g. PhD, MD	
Occupation	3	What was their occupation at the time of the study?	
Gender	4	Was the researcher male or female?	
Experience and training	5	What experience or training did the researcher have?	
Relationship with participants			
Relationship established	6	Was a relationship established prior to study commencement?	
Participant knowledge of the interviewer	7	What did the participants know about the researcher? e.g. personal goals, reasons for doing the research	
Interviewer characteristics	8	What characteristics were reported about the interviewer/facilitator? e.g. Bias, assumptions, reasons and interests in the research topic	
Domain 2: Study design			
Theoretical framework			
Methodological orientation and Theory	9	What methodological orientation was stated to underpin the study? e.g. grounded theory, discourse analysis, ethnography, phenomenology, content analysis	
Participant selection			
Sampling	10	How were participants selected? e.g. purposive, convenience, consecutive, snowball	
Method of approach	11	How were participants approached? e.g. face-to-face, telephone, mail, email	
Sample size	12	How many participants were in the study?	
Non-participation	13	How many people refused to participate or dropped out? Reasons?	
Setting			
Setting of data collection	14	Where was the data collected? e.g. home, clinic, workplace	
Presence of non-participants	15	Was anyone else present besides the participants and researchers?	
Description of sample	16	What are the important characteristics of the sample? e.g. demographic data, date	
Data collection			
Interview guide	17	Were questions, prompts, guides provided by the authors? Was it pilot tested?	
Repeat interviews	18	Were repeat interviews carried out? If yes, how many?	
Audio/visual recording	19	Did the research use audio or visual recording to collect the data?	
Field notes	20	Were field notes made during and/or after the interview or focus group?	
Duration	21	What was the duration of the interviews or focus group?	
Data saturation	22	Was data saturation discussed?	
Transcripts returned	23	Were transcripts returned to participants for comment and/or	

Topic	Item No.	Guide Questions/Description	Reported on Page No.
		correction?	
Domain 3: analysis and findings			
Data analysis			
Number of data coders	24	How many data coders coded the data?	
Description of the coding tree	25	Did authors provide a description of the coding tree?	
Derivation of themes	26	Were themes identified in advance or derived from the data?	
Software	27	What software, if applicable, was used to manage the data?	
Participant checking	28	Did participants provide feedback on the findings?	
Reporting			
Quotations presented	29	Were participant quotations presented to illustrate the themes/findings? Was each quotation identified? e.g. participant number	
Data and findings consistent	30	Was there consistency between the data presented and the findings?	
Clarity of major themes	31	Were major themes clearly presented in the findings?	
Clarity of minor themes	32	Is there a description of diverse cases or discussion of minor themes?	

Developed from: Tong A, Sainsbury P, Craig J. Consolidated criteria for reporting qualitative research (COREQ): a 32-item checklist for interviews and focus groups. *International Journal for Quality in Health Care*. 2007. Volume 19, Number 6: pp. 349 – 357

Once you have completed this checklist, please save a copy and upload it as part of your submission. DO NOT include this checklist as part of the main manuscript document. It must be uploaded as a separate file.

BMJ Open

What matters to medical ward patients, and do we measure it?

A qualitative comparison of patient priorities and current practice in quality measurement, on UK NHS medical wards

Journal:	BMJ Open
Manuscript ID	bmjopen-2018-024058.R1
Article Type:	Research
Date Submitted by the Author:	19-Sep-2018
Complete List of Authors:	Pannick, Samuel; Charing Cross Hospital, Gastroenterology Archer, Stephanie; Imperial College London, NIHR Imperial Patient Safety Translational Research Centre; Imperial College London, Long, Susannah; Imperial College London, Centre for Patient Safety and Service Quality Husson, Fran; Imperial College, Faculty of Medicine Athanasiou, Thanos; Imperial College London, Surgery and Cancer Sevdalis, Nick; King's College London,
Primary Subject Heading:	Health services research
Secondary Subject Heading:	Evidence based practice, Patient-centred medicine, Qualitative research
Keywords:	Quality in health care < HEALTH SERVICES ADMINISTRATION & MANAGEMENT, GENERAL MEDICINE (see Internal Medicine), Organisation of health services < HEALTH SERVICES ADMINISTRATION & MANAGEMENT, Clinical governance < HEALTH SERVICES ADMINISTRATION & MANAGEMENT

**What matters to medical ward patients, and do we measure it?**
**A qualitative comparison of patient priorities and current practice**
**in quality measurement, on UK NHS medical wards**

Samuel Pannick,^{1,2} Stephanie Archer,¹ Susannah J Long,^{1,2} Fran Husson,^{1,2}
Thanos Athanasiou,³ & Nick Sevdalis⁴

¹ NIHR Imperial Patient Safety Translational Research Centre, Imperial College London, UK

² Imperial College Healthcare NHS Trust, London, UK

³ Department of Surgery & Cancer, Imperial College London, UK

⁴ Centre for Implementation Science, King's College London, UK

Corresponding author:

Dr Sam Pannick, MA MB BS MRCP PhD

Specialist Registrar in Gastroenterology and Internal Medicine, Department of

Gastroenterology, Charing Cross Hospital, Fulham Palace Road, W6 8RF

United Kingdom

sam.pannick@nhs.net; + (44) 203 311 1234

*Author information:*

Dr Stephanie Archer, Research Fellow, NIHR Imperial Patient Safety Translational Research
Centre, Imperial College London.

Dr Susannah J Long, Consultant Geriatrician, Imperial College Healthcare NHS Trust, and
Honorary Clinical Senior Lecturer, NIHR Imperial Patient Safety Translational Research
Centre, Imperial College London.

Ms Fran Husson, Patient Advocate, Imperial College Healthcare NHS Trust, and Honorary
Research Officer, Imperial College London.

Professor Thanos Athanasiou, Consultant Cardiothoracic Surgeon, Imperial College
Healthcare NHS Trust, and Professor of Cardiovascular Sciences and Cardiac Surgery,
Imperial College London.

Professor Nick Sevdalis, Professor of Implementation Science and Patient Safety, King's
College London.

*Word count: 3133*

*Conflicts of interest:* This paper represents independent research supported by the National
Institute for Health Research (NIHR) Imperial Patient Safety Translational Research Centre,
and the Imperial College Healthcare Charity (Grant GG14\1022). NS' research is supported
by the NIHR Collaboration for Leadership in Applied Health Research and Care South London
at King's College Hospital NHS Foundation Trust. NS is a member of King's Improvement
Science, which is part of the NIHR CLAHRC South London and comprises a specialist team of
improvement scientists and senior researchers based at King's College London. Its work is
funded by King's Health Partners (Guy's and St Thomas' NHS Foundation Trust, King's
College Hospital NHS Foundation Trust, King's College London and South London and

Maudsley NHS Foundation Trust), Guy's and St Thomas' Charity, the Maudsley Charity and
the Health Foundation. No funding source had any role in the design and conduct of the
study; collection, management, analysis or interpretation of the data; or preparation, review
or approval of the manuscript. The views expressed are those of the authors and not
necessarily those of the NHS, the NIHR or the Department of Health and Social Care.

*Contributions:*

Study design: SP, SA, TA, FH, NS, SJL. Study implementation and data collection: SA, SP, SJL.
Analysis: SA, SP, NS, FH, TA. All authors contributed to, read and approved the final
manuscript.

*Ethical approval:*

Ethical approval was granted by the Westminster Research Ethics Committee (16/LO/0196)
and the hospital's joint research compliance office (16SM3129).

*Acknowledgements:* We are grateful to Drs Louise Hull and Tayana Soukup for their help in
facilitating the focus groups. We thank Ms Margaret Turley (carer representative) from the
NIHR Imperial PSTRC for her help developing the interview guide.

*Data sharing:* There are no additional unpublished data.

*Key words:* healthcare quality; medical ward; patient experience

[revised manuscript text omitted]

TOPIC GUIDE: Current patient / carer for current patient**1. Let's find out about you.**

Age
Employment
Education – school / university / postgraduate
Social support structures & marital status
Ethnicity

2. Why are you in hospital now?

Current diagnosis
Other conditions
Approximate length of stay to date
 < 1 day
 1 – 5 days
 5 – 10 days
 > 10 days

3. How many times have you been admitted to hospital in the last 6 months?

1-5
5-10
>10

4. Do you always come to this hospital or have you been admitted to other local hospitals?**5. How do you know if you're on a good ward? What is a 'good ward' to you?**

Environment

- Clean
- Quiet
- Toilet and shower are available when required
- Meal timeliness, warmth
- Help available when requested
- Staff are responsiveness to my needs / my family's needs

Welcome

- My arrival is expected
- Staff introduce themselves
- Staff make me feel I will be well looked after; show a caring attitude; and don't rush me

Communication and use of personal information

- Accurate knowledge of previous medical history / current diagnosis / current investigations / discharge plan / medication reconciliation
- Quality of communication / teamwork

Discharge preparation

Friends' / families' recommendations

Ward information boards / quality and safety boards

Ward information leaflets / other printed materials.

Ward information displays / electronic screens

**6. If you had to decide whether a ward was good or not, what information would you need to**
**make that decision?**

**7. Have you noticed any of the information the ward displays about itself? What do you think of**
**the information you've seen?**

Friends and family test results

Safety cross

Shift-by-shift staffing

Falls

Pressure ulcers

Safety thermometer / harm-free care

Venous thromboembolism prophylaxis

Hand hygiene compliance

Hospital-acquired infections

Infection rates

Incident reporting

**8. What would you like to know about how your ward is performing?**

Hand hygiene compliance

Staffing levels

Friends and family results

Hospital-acquired infections

Pressure Ulcers

Falls

Venous thromboembolism prophylaxis

Complaints

Compliments

Length of stay

Mortality

Readmission rate

Safety climate

**9. How should your ward make that information available to you and your family??**

Ward displays

Leaflets

Smartphone / other device

Webpage

**10. Preference for information seeking**

Information-seeking sub-scale

	Disagree strongly	Disagree slightly	Neutral	Agree slightly	Agree strongly
11 12 As you become sicker you should be told more and more about your illness
13 14 15 You should understand completely what is happening inside your body as a result of your illness					
16 17 18 Even if the news is bad, you should be well informed					
19 20 21 Your doctor should explain the purpose of your laboratory tests					
22 23 24 It is important for you to know all the side effects of your medication					
25 26 27 Information about your illness is as important to you as treatment					
28 29 30 When there is more than one method to treat a problem, you should be told about each one					

**11. Have you previously had to complain about care or healthcare staff, nurses or doctors? What made you complain? How? PALS / informally / in writing?**

Ward: [redacted]
Patient Safety Information

Our last pressure ulcer (graded 1-4) was on:

JULY 2015

Our last patient fall was on:

23.9.2015

Ward: [redacted]
Infection Control Information

Our last incidence of MRSA was on:

25.05.14

Our last incidence of C.Diff was on:

15.12.14

I-Staff Safe Staffing Information Board

The Nurse in Charge today is:

[Redacted]

Staffing Status

[Redacted]

DATE:

15-2-15

Current Shift

Registered Nurses

Staff required

On duty

3

1

Healthcare Assistants

2

2

Additional Support
(for e.g. specials)

Roles and Responsibilities

	Responsibilities
Nurse in Charge	Provides supervision, management and direct clinical care
Registered Nurses	Provides direct clinical care and supervision
Health Care Assistants	Provides direct care under the supervision of a registered nurse
Additional Support	Specials (may be registered or unregistered providing 121 direct care with supervision)

If you have any queries about the staffing on this ward please speak to the Nurse in Charge

COREQ (CONsolidated criteria for REporting Qualitative research) Checklist

A checklist of items that should be included in reports of qualitative research. You must report the page number in your manuscript where you consider each of the items listed in this checklist. If you have not included this information, either revise your manuscript accordingly before submitting or note N/A.

Topic	Item No.	Guide Questions/Description	Reported on Page No.
Domain 1: Research team and reflexivity			
Personal characteristics			
Interviewer/facilitator	1	Which author/s conducted the interview or focus group?	
Credentials	2	What were the researcher's credentials? E.g. PhD, MD	
Occupation	3	What was their occupation at the time of the study?	
Gender	4	Was the researcher male or female?	
Experience and training	5	What experience or training did the researcher have?	
Relationship with participants			
Relationship established	6	Was a relationship established prior to study commencement?	
Participant knowledge of the interviewer	7	What did the participants know about the researcher? e.g. personal goals, reasons for doing the research	
Interviewer characteristics	8	What characteristics were reported about the interviewer/facilitator? e.g. Bias, assumptions, reasons and interests in the research topic	
Domain 2: Study design			
Theoretical framework			
Methodological orientation and Theory	9	What methodological orientation was stated to underpin the study? e.g. grounded theory, discourse analysis, ethnography, phenomenology, content analysis	
Participant selection			
Sampling	10	How were participants selected? e.g. purposive, convenience, consecutive, snowball	
Method of approach	11	How were participants approached? e.g. face-to-face, telephone, mail, email	
Sample size	12	How many participants were in the study?	
Non-participation	13	How many people refused to participate or dropped out? Reasons?	
Setting			
Setting of data collection	14	Where was the data collected? e.g. home, clinic, workplace	
Presence of non-participants	15	Was anyone else present besides the participants and researchers?	
Description of sample	16	What are the important characteristics of the sample? e.g. demographic data, date	
Data collection			
Interview guide	17	Were questions, prompts, guides provided by the authors? Was it pilot tested?	
Repeat interviews	18	Were repeat interviews carried out? If yes, how many?	
Audio/visual recording	19	Did the research use audio or visual recording to collect the data?	
Field notes	20	Were field notes made during and/or after the interview or focus group?	
Duration	21	What was the duration of the interviews or focus group?	
Data saturation	22	Was data saturation discussed?	
Transcripts returned	23	Were transcripts returned to participants for comment and/or	

Topic	Item No.	Guide Questions/Description	Reported on Page No.
		correction?	
Domain 3: analysis and findings			
Data analysis			
Number of data coders	24	How many data coders coded the data?	
Description of the coding tree	25	Did authors provide a description of the coding tree?	
Derivation of themes	26	Were themes identified in advance or derived from the data?	
Software	27	What software, if applicable, was used to manage the data?	
Participant checking	28	Did participants provide feedback on the findings?	
Reporting			
Quotations presented	29	Were participant quotations presented to illustrate the themes/findings? Was each quotation identified? e.g. participant number	
Data and findings consistent	30	Was there consistency between the data presented and the findings?	
Clarity of major themes	31	Were major themes clearly presented in the findings?	
Clarity of minor themes	32	Is there a description of diverse cases or discussion of minor themes?	

Developed from: Tong A, Sainsbury P, Craig J. Consolidated criteria for reporting qualitative research (COREQ): a 32-item checklist for interviews and focus groups. *International Journal for Quality in Health Care*. 2007. Volume 19, Number 6: pp. 349 – 357

Once you have completed this checklist, please save a copy and upload it as part of your submission. DO NOT include this checklist as part of the main manuscript document. It must be uploaded as a separate file.

BMJ Open

What matters to medical ward patients, and do we measure it?

A qualitative comparison of patient priorities and current practice in quality measurement, on UK NHS medical wards

Journal:	BMJ Open
Manuscript ID	bmjopen-2018-024058.R2
Article Type:	Research
Date Submitted by the Author:	18-Dec-2018
Complete List of Authors:	Pannick, Samuel; Charing Cross Hospital, Gastroenterology Archer, Stephanie; Imperial College London, NIHR Imperial Patient Safety Translational Research Centre; Imperial College London, Long, Susannah; Imperial College London, Centre for Patient Safety and Service Quality Husson, Fran; Imperial College, Faculty of Medicine Athanasiou, Thanos; Imperial College London, Surgery and Cancer Sevdalis, Nick; King's College London,
Primary Subject Heading:	Health services research
Secondary Subject Heading:	Evidence based practice, Patient-centred medicine, Qualitative research
Keywords:	Quality in health care < HEALTH SERVICES ADMINISTRATION & MANAGEMENT, GENERAL MEDICINE (see Internal Medicine), Organisation of health services < HEALTH SERVICES ADMINISTRATION & MANAGEMENT, Clinical governance < HEALTH SERVICES ADMINISTRATION & MANAGEMENT

What matters to medical ward patients, and do we measure it?

A qualitative comparison of patient priorities and current practice in quality measurement, on UK NHS medical wards

Samuel Pannick,^{1,2} Stephanie Archer,¹ Susannah J Long,^{1,2} Fran Husson,^{1,2}

Thanos Athanasiou,³ & Nick Sevdalis⁴

¹ NIHR Imperial Patient Safety Translational Research Centre, Imperial College London, UK

² Imperial College Healthcare NHS Trust, London, UK

³ Department of Surgery & Cancer, Imperial College London, UK

⁴ Centre for Implementation Science, King's College London, UK

Corresponding author:

Dr Sam Pannick, MA MB BS MRCP PhD

Specialist Registrar in Gastroenterology and Internal Medicine, Department of

Gastroenterology, Charing Cross Hospital, Fulham Palace Road, W6 8RF

United Kingdom

sam.pannick@nhs.net; + (44) 203 311 1234

Author information:

Dr Stephanie Archer, Research Fellow, NIHR Imperial Patient Safety Translational Research Centre, Imperial College London.

Dr Susannah J Long, Consultant Geriatrician, Imperial College Healthcare NHS Trust, and Honorary Clinical Senior Lecturer, NIHR Imperial Patient Safety Translational Research Centre, Imperial College London.

Ms Fran Husson, Patient Advocate, Imperial College Healthcare NHS Trust, and Honorary Research Officer, Imperial College London.

Professor Thanos Athanasiou, Consultant Cardiothoracic Surgeon, Imperial College Healthcare NHS Trust, and Professor of Cardiovascular Sciences and Cardiac Surgery, Imperial College London.

Professor Nick Sevdalis, Professor of Implementation Science and Patient Safety, King's College London.

Conflicts of interest: This paper represents independent research supported by the National Institute for Health Research (NIHR) Imperial Patient Safety Translational Research Centre, and the Imperial College Healthcare Charity (Grant GG14\1022). NS' research is supported by the NIHR Collaboration for Leadership in Applied Health Research and Care South London at King's College Hospital NHS Foundation Trust. NS is a member of King's Improvement Science, which is part of the NIHR CLAHRC South London and comprises a specialist team of improvement scientists and senior researchers based at King's College London. Its work is funded by King's Health Partners (Guy's and St Thomas' NHS Foundation Trust, King's College Hospital NHS Foundation Trust, King's College London and South London and Maudsley NHS Foundation Trust), Guy's and St Thomas' Charity, the Maudsley Charity and the Health Foundation. No funding source had any role in the design and conduct of the

study; collection, management, analysis or interpretation of the data; or preparation, review
or approval of the manuscript. The views expressed are those of the authors and not
necessarily those of the NHS, the NIHR or the Department of Health and Social Care.

*Contributions:*

Study design: SP, SA, TA, FH, NS, SJL. Study implementation and data collection: SA, SP, SJL.

Analysis: SA, SP, NS, FH, TA. All authors contributed to, read and approved the final
manuscript.

*Ethical approval:*

Ethical approval was granted by the Westminster Research Ethics Committee (16/LO/0196)
and the hospital's joint research compliance office (16SM3129).

*Acknowledgements:* We are grateful to Drs Louise Hull and Tayana Soukup for their help in
facilitating the focus groups. We thank Ms Margaret Turley (carer representative) from the
NIHR Imperial PSTRC for her help developing the interview guide.

*Data sharing:* There are no additional unpublished data.

*Key words:* healthcare quality; medical ward; patient experience

[revised manuscript text omitted]

TOPIC GUIDE: Current patient / carer for current patient**1. Let's find out about you.**

Age
Employment
Education – school / university / postgraduate
Social support structures & marital status
Ethnicity

2. Why are you in hospital now?

Current diagnosis
Other conditions
Approximate length of stay to date
 < 1 day
 1 – 5 days
 5 – 10 days
 > 10 days

3. How many times have you been admitted to hospital in the last 6 months?

1-5
5-10
>10

4. Do you always come to this hospital or have you been admitted to other local hospitals?**5. How do you know if you're on a good ward? What is a 'good ward' to you?**

Environment

- Clean
- Quiet
- Toilet and shower are available when required
- Meal timeliness, warmth
- Help available when requested
- Staff are responsiveness to my needs / my family's needs

Welcome

- My arrival is expected
- Staff introduce themselves
- Staff make me feel I will be well looked after; show a caring attitude; and don't rush me

Communication and use of personal information

- Accurate knowledge of previous medical history / current diagnosis / current investigations / discharge plan / medication reconciliation
- Quality of communication / teamwork

Discharge preparation

Friends' / families' recommendations

Ward information boards / quality and safety boards

Ward information leaflets / other printed materials.

Ward information displays / electronic screens

**6. If you had to decide whether a ward was good or not, what information would you need to**
**make that decision?**

**7. Have you noticed any of the information the ward displays about itself? What do you think of**
**the information you've seen?**

Friends and family test results

Safety cross

Shift-by-shift staffing

Falls

Pressure ulcers

Safety thermometer / harm-free care

Venous thromboembolism prophylaxis

Hand hygiene compliance

Hospital-acquired infections

Infection rates

Incident reporting

**8. What would you like to know about how your ward is performing?**

Hand hygiene compliance

Staffing levels

Friends and family results

Hospital-acquired infections

Pressure Ulcers

Falls

Venous thromboembolism prophylaxis

Complaints

Compliments

Length of stay

Mortality

Readmission rate

Safety climate

**9. How should your ward make that information available to you and your family??**

Ward displays

Leaflets

Smartphone / other device

Webpage

**10. Preference for information seeking**

Information-seeking sub-scale

	Disagree strongly	Disagree slightly	Neutral	Agree slightly	Agree strongly
11 12 As you become sicker you should be told more and more about your illness
13 14 15 You should understand completely what is happening inside your body as a result of your illness					
16 17 18 Even if the news is bad, you should be well informed					
19 20 21 Your doctor should explain the purpose of your laboratory tests					
22 23 24 It is important for you to know all the side effects of your medication					
25 26 27 Information about your illness is as important to you as treatment					
28 29 30 When there is more than one method to treat a problem, you should be told about each one					

**11. Have you previously had to complain about care or healthcare staff, nurses or doctors? What made you complain? How? PALS / informally / in writing?**

I-Staff Safe Staffing Information Board

The Nurse in Charge today is:

[Redacted]

Staffing Status

[Redacted]

DATE:

15-2-15

Current Shift

Registered Nurses

Staff required

On duty

3

1

Healthcare Assistants

2

2

Additional Support
(for e.g. specials)

Roles and Responsibilities

	Responsibilities
Nurse in Charge	Provides supervision, management and direct clinical care
Registered Nurses	Provides direct clinical care and supervision
Health Care Assistants	Provides direct care under the supervision of a registered nurse
Additional Support	Specials (may be registered or unregistered providing 121 direct care with supervision)

If you have any queries about the staffing on this ward please speak to the Nurse in Charge

COREQ (COnsolidated criteria for REporting Qualitative research) Checklist

A checklist of items that should be included in reports of qualitative research. You must report the page number in your manuscript where you consider each of the items listed in this checklist. If you have not included this information, either revise your manuscript accordingly before submitting or note N/A.

Topic	Item No.	Guide Questions/Description	Reported on Page No.
Domain 1: Research team and reflexivity			
Personal characteristics			
Interviewer/facilitator	1	Which author/s conducted the interview or focus group?	
Credentials	2	What were the researcher's credentials? E.g. PhD, MD	
Occupation	3	What was their occupation at the time of the study?	
Gender	4	Was the researcher male or female?	
Experience and training	5	What experience or training did the researcher have?	
Relationship with participants			
Relationship established	6	Was a relationship established prior to study commencement?	
Participant knowledge of the interviewer	7	What did the participants know about the researcher? e.g. personal goals, reasons for doing the research	
Interviewer characteristics	8	What characteristics were reported about the interviewer/facilitator? e.g. Bias, assumptions, reasons and interests in the research topic	
Domain 2: Study design			
Theoretical framework			
Methodological orientation and Theory	9	What methodological orientation was stated to underpin the study? e.g. grounded theory, discourse analysis, ethnography, phenomenology, content analysis	
Participant selection			
Sampling	10	How were participants selected? e.g. purposive, convenience, consecutive, snowball	
Method of approach	11	How were participants approached? e.g. face-to-face, telephone, mail, email	
Sample size	12	How many participants were in the study?	
Non-participation	13	How many people refused to participate or dropped out? Reasons?	
Setting			
Setting of data collection	14	Where was the data collected? e.g. home, clinic, workplace	
Presence of non-participants	15	Was anyone else present besides the participants and researchers?	
Description of sample	16	What are the important characteristics of the sample? e.g. demographic data, date	
Data collection			
Interview guide	17	Were questions, prompts, guides provided by the authors? Was it pilot tested?	
Repeat interviews	18	Were repeat interviews carried out? If yes, how many?	
Audio/visual recording	19	Did the research use audio or visual recording to collect the data?	
Field notes	20	Were field notes made during and/or after the interview or focus group?	
Duration	21	What was the duration of the interviews or focus group?	
Data saturation	22	Was data saturation discussed?	
Transcripts returned	23	Were transcripts returned to participants for comment and/or	

Topic	Item No.	Guide Questions/Description	Reported on Page No.
		correction?	
Domain 3: analysis and findings			
Data analysis			
Number of data coders	24	How many data coders coded the data?	
Description of the coding tree	25	Did authors provide a description of the coding tree?	
Derivation of themes	26	Were themes identified in advance or derived from the data?	
Software	27	What software, if applicable, was used to manage the data?	
Participant checking	28	Did participants provide feedback on the findings?	
Reporting			
Quotations presented	29	Were participant quotations presented to illustrate the themes/findings? Was each quotation identified? e.g. participant number	
Data and findings consistent	30	Was there consistency between the data presented and the findings?	
Clarity of major themes	31	Were major themes clearly presented in the findings?	
Clarity of minor themes	32	Is there a description of diverse cases or discussion of minor themes?	

Developed from: Tong A, Sainsbury P, Craig J. Consolidated criteria for reporting qualitative research (COREQ): a 32-item checklist for interviews and focus groups. *International Journal for Quality in Health Care*. 2007. Volume 19, Number 6: pp. 349 – 357

Once you have completed this checklist, please save a copy and upload it as part of your submission. DO NOT include this checklist as part of the main manuscript document. It must be uploaded as a separate file.